# Injecting Textual Spatial Context into Vision-Language Models for Surgical Scene Understanding

**Antonio Martignano**[1]                    ANTONIO.MARTIGNANO@SLU.EDU

**Lin Guo**[1]                                     LIN.GUO@SLU.EDU

**Chiara Camerota**[1]                    CHIARA.CAMEROTA@SLU.EDU

[1] *Computer Science Department, Saint Louis University, USA*

**Alessio Sacco**[2]                         ALESSIO_SACCO@POLITO.IT

[2] *Department of Control and Computer Engineering, Politecnico di Torino, Italy*

**Flavio Esposito**[1]                       FLAVIO.ESPOSITO@SLU.EDU

## Abstract

Accurate anatomical landmark identification is important for safe laparoscopic navigation, yet limited view and strong tissue similarity make multi-label organ classification difficult. Existing vision-language models mainly rely on appearance and overlook the spatial structure of surgical scenes (Zhang et al., 2025). We propose *SpatialContext*, a multi-modal framework that injects scene geometry into classification through natural language prompts derived from segmentation masks, together with a context-conditional training strategy centered on the primary surgical target. Results on DSAD (Carstens et al., 2023) and Endoscapes (Mascagni et al., 2025) show improved recognition of scene-defining and off-target anatomy, suggesting that explicit spatial semantics can improve surgical scene understanding.

**Keywords:** Laparoscopic Surgery, Vision-Language Models, Spatial Awareness.

## 1. Introduction

Laparoscopic surgeons operate from a restricted two-dimensional view, where visually similar tissues can make critical anatomy difficult to identify and increase the risk of iatrogenic injury (Kim et al., 2023). Although computer vision has strong potential as an intraoperative guide (Maier-Hein et al., 2022), dense supervision remains expensive in surgery (Willemink et al., 2020). As a result, datasets such as DSAD (Carstens et al., 2023) provide one dense primary-organ mask per image together with weak labels for additional visible structures, enabling scalable but challenging multi-label recognition. Standard detectors based on bounding boxes, such as YOLO (Redmon et al., 2016), are often too coarse for irregular anatomy, while weakly supervised classifiers still struggle with strong inter-tissue similarity.

Vision-language models such as CLIP (Radford et al., 2021) and BiomedCLIP (Zhang et al., 2025) offer an appealing alternative by aligning images and text without requiring closed-set classifiers, and BiomedCoOp (Koleilat et al., 2025) further improves this paradigm through learnable biomedical prompts. However, these methods primarily rely on visual appearance and do not explicitly model the spatial organization of surgical scenes. Graph-based approaches attempt to encode anatomy through scene relationships (Yuan et al., 2024), but they depend on reliable object detection and can fail when key structures are not detected. We therefore propose *SpatialContext*, an initial step toward injecting explicit spatial cues into surgical vision-language modeling through natural language derived from the primary organ mask, with the broader goal of moving from texture matching toward scene-aware anatomical understanding.

## 2. System Design

Figure 1 illustrates SpatialContext. Frozen BiomedCLIP image and text encoders extract visual features and encode a spatial prompt generated from the centroid of the primary organ mask (e.g., *"The liver is in the top-right"*). Their embeddings are fused by an MLP and passed to independent sigmoid heads trained with weighted binary cross-entropy, enabling multi-label prediction. To avoid oracle leakage, context-conditional training masks the primary organ from both predictions and labels, forcing the model to use it only as a spatial anchor for detecting secondary anatomy.

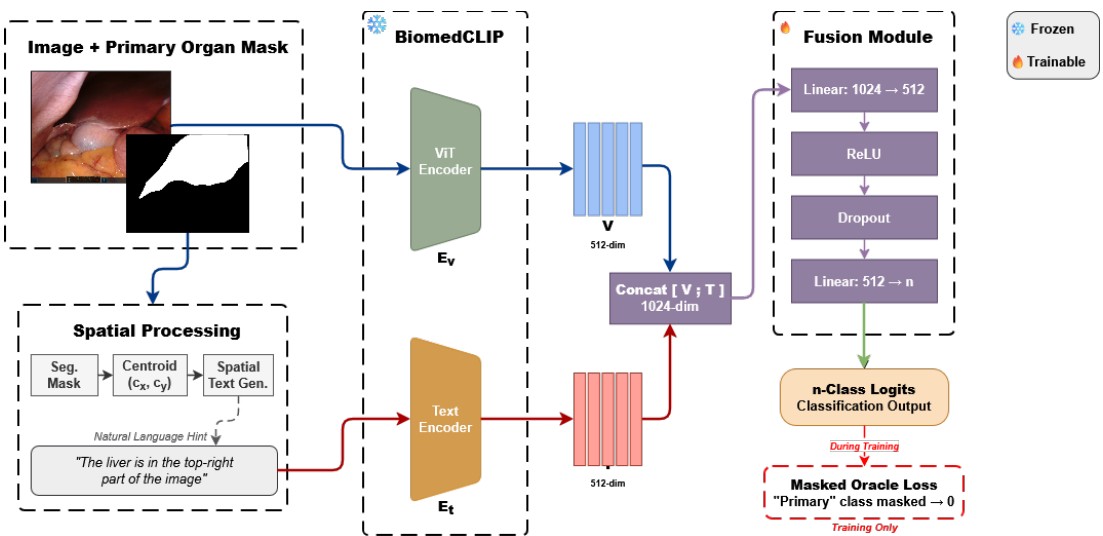

Figure 1: Overview of SpatialContext architecture.

## 3. Experiment and Results

We compare SpatialContext with zero-shot BiomedCLIP, BiomedCoOp, and a supervised YOLOv9 (Wang et al., 2025) baseline on DSAD and Endoscapes. To address class imbalance, we weight the positive term of binary cross-entropy by the class-specific ratio of negative to positive samples. We also sweep decision thresholds $\tau \in [0.001, 0.99]$ to distinguish detection errors from calibration errors.

**Global performance.** Table 1 shows that SpatialContext achieves 27.90% mAP on DSAD, outperforming BiomedCoOp, YOLOv9, and zero-shot BiomedCLIP. Figure 2 further shows that its gains are concentrated on large scene-defining organs such as the *Abdominal Wall* and *Small Intestine*, with additional improvement on the *Pancreas* and *Liver*, whereas BiomedCoOp remains stronger on finer localized structures including the *Inferior Mesenteric Artery (IMA)*, *Intestinal Veins*, and *Stomach*, suggesting that coarse spatial prompting mainly improves global scene understanding while prompt tuning better captures fine-grained local appearance.

**Scene vs. structure trade-off.** Spatial priors dominate on large, texture-poor regions that define the macroscopic scene: Abdominal Wall (AP 82.3%) and Small Intestine (AP 77.1%). For compact micro-anatomy defined by fine texture: ureter (AP 4.6%), spleen (AP 0.8%), BiomedCoOp retains superior performance. Spatial prompts introduce

Table 1: Global performance under masked evaluation.

| Dataset | Model | mAP | F1 ($\tau = 0.5$) | Opt. F1 |
|---|---|---|---|---|
| DSAD (11 classes) | SpatialContext | **27.90%** | **45.23%** | **50.99%** |
| | BiomedCoOp | 25.43% | 15.23% | 30.79% |
| | YOLOv9 (Expert) | 13.15% | 0.82% | 23.47% |
| | BiomedCLIP (ZS) | 12.64% | 19.47% | 19.47% |
| Endoscapes (5 classes) | SpatialContext | 72.56% | 27.54% | **80.89%** |
| | BiomedCoOp | **75.16%** | 0.20% | 80.45% |
| | YOLOv9 (Expert) | 69.39% | 10.47% | 70.93% |
| | BiomedCLIP (ZS) | 69.07% | **79.54%** | 79.54% |

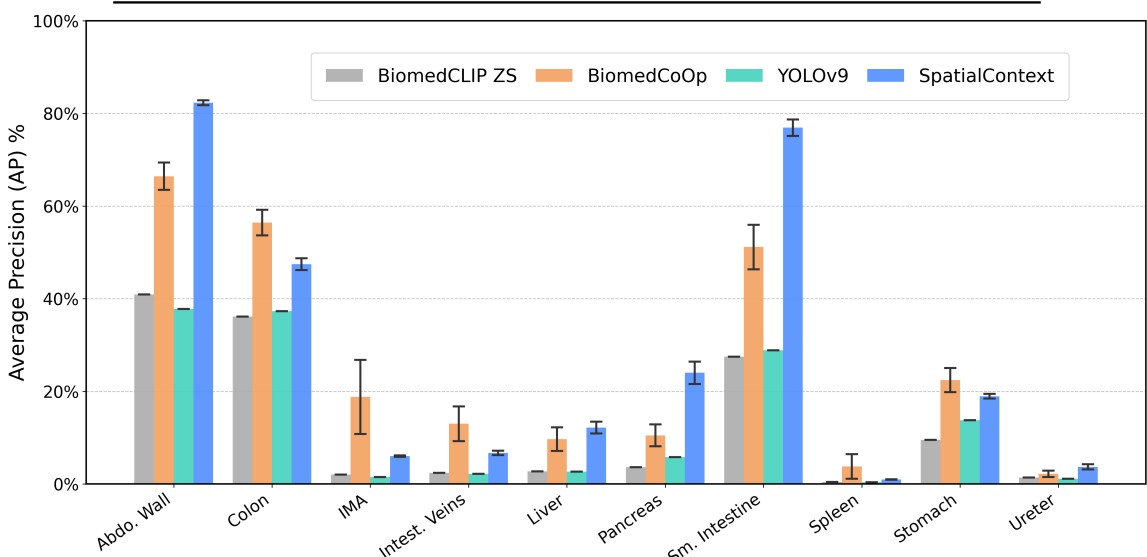

Figure 2: Per-class AP comparison on DSAD under masked evaluation.

a low-resolution scene prior that regularises toward global layout at the cost of local detail. Prompt tuning conversely excels at high-resolution texture matching.

## 4. Conclusion

SpatialContext converts geometric centroids into natural language prompts, injecting coarse spatial context into a frozen VLM without dense graph annotation. Context-Conditional Training ensures genuine multimodal learning, while post-training threshold calibration remains important under severe class imbalance. Overall, this study provides an initial demonstration that coarse spatial prompting can improve scene-level anatomical understanding in weakly supervised surgical images. Future work will incorporate finer spatial detail and explicit inter-organ relations for more accurate spatial reasoning.

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
