# OpenReview forum: "Injecting Textual Spatial Context into Vision-Language Models for Surgical Scene Understanding"
_MIDL.io/2026/Short_Papers — MIDL 2026 - Short Papers Poster_

### Official Review · Reviewer_V9Zj · 2026-04-22
**Coarse spatial prompting can improve scene-level anatomical understanding in surgical images**

**Rating:** 4
**Confidence:** 4

**Review:**

Overall, this is a good paper. The idea of injecting spatial prompts into a vision-language model to improve scene-level anatomical understanding in surgical images is novel. The comparison with BiomedCoOp further elucidates the limitation of coarse spatial prompts and a promising direction of incorporating finer spatial detail and explicit inter-organ relationships for more accurate spatial reasoning in surgical images.

**Summary:**

The paper propose to use coarse spatial prompting generated from geometric centroids of segmentation masks to improve scene-level anatomical understanding in surgical images. The authors analyzed the reasons that lead the performance difference between the proposed method and a strong baseline method BiomedCoOp.

**Strengths:**

1. The paper is well-written.
2. The idea of injecting spatial prompts into a vision-language model to improve scene-level anatomical understanding in surgical images is novel.
3. The authors included two strong baseline methods for comparison.
4. The comparison with BiomedCoOp further elucidates the limitation of coarse spatial prompts and a promising direction of incorporating finer spatial detail and explicit inter-organ relationships for more accurate spatial reasoning in surgical images.

**Weaknesses:**

1. It remains unclear how to compute the masked oracle loss.
2. Will it be better to assign a weight to the primary organ instead of completely masking out it?
3. In addition to use the geometric centroids of segmentation masks to form a spatial prompt, can we derive out more features from the image and the mask to form a more sophisticated prompt? E.g., The shape of the liver, the spatial range of the liver, the pixel color of the liver, etc.

**Justification Of Rating:**

The idea of injecting spatial prompts into a vision-language model to improve scene-level anatomical understanding is novel. The comparison with BiomedCoOp further elucidates the limitation of the proposed method and a promising direction for more accurate spatial reasoning in surgical images.

---

### Decision · Program_Chairs · 2026-05-08

Accept (Poster)